# Biomarkers of Neurological Damage: From Acute Stage to Post-Acute Sequelae of COVID-19

**DOI:** 10.3390/cells12182270

**Published:** 2023-09-13

**Authors:** Maria Antonella Zingaropoli, Patrizia Pasculli, Christian Barbato, Carla Petrella, Marco Fiore, Federica Dominelli, Tiziana Latronico, Federica Ciccone, Michele Antonacci, Grazia Maria Liuzzi, Giuseppina Talarico, Giuseppe Bruno, Gioacchino Galardo, Francesco Pugliese, Miriam Lichtner, Claudio Maria Mastroianni, Antonio Minni, Maria Rosa Ciardi

**Affiliations:** 1Department of Public Health and Infectious Diseases, Sapienza University of Rome, Piazzale Aldo Moro 5, 00185 Rome, Italy; patrizia.pasculli@uniroma1.it (P.P.); federica.dominelli@uniroma1.it (F.D.); federica.ciccone@uniroma1.it (F.C.); michele.anto26@gmail.com (M.A.); claudio.mastroianni@uniroma1.it (C.M.M.); maria.ciardi@uniroma1.it (M.R.C.); 2Department of Sense Organs, Institute of Biochemistry and Cell Biology (IBBC), National Research Council (CNR), Sapienza University of Rome, 00185 Rome, Italy; christian.barbato@cnr.it (C.B.); carla.petrella@cnr.it (C.P.); marco.fiore@cnr.it (M.F.); 3Department of Biosciences, Biotechnologies and Environment, University of Bari Aldo Moro, 70121 Bari, Italy; tiziana.latronico@uniba.it (T.L.); graziamaria.liuzzi@uniba.it (G.M.L.); 4Department of Human Neuroscience, Sapienza University of Rome, 00185 Rome, Italy; giuseppina.talarico@uniroma1.it (G.T.); giuseppe.bruno@uniroma1.it (G.B.); 5Medical Emergency Unit, Sapienza University of Rome, Policlinico Umberto I, 00161 Rome, Italy; gioacchino.galardo@uniroma1.it; 6Department of Specialist Surgery and Organ Transplantation “Paride Stefanini”, Policlinico Umberto I, Sapienza University of Rome, 00161 Rome, Italy; f.pugliese@uniroma1.it; 7Infectious Diseases Unit, SM Goretti Hospital, Sapienza University of Rome, 00185 Latina, Italy; miriam.lichtner@uniroma1.it; 8Department of Neurosciences, Mental Health, and Sense Organs, Sapienza University of Rome, 00185 Rome, Italy; 9Department of Sensory Organs, Sapienza University of Rome, 00185 Rome, Italy; antonio.minni@uniroma1.it; 10Division of Otolaryngology-Head and Neck Surgery, ASL Rieti-Sapienza University, Ospedale San Camillo de Lellis, Viale Kennedy, 02100 Rieti, Italy

**Keywords:** neurofilament light chain, glial fibrillary acidic protein, central nervous system, cerebrospinal fluid, long-COVID, neuro-COVID, NfL, GFAP, sCD163

## Abstract

**Background**: Neurological symptoms (NS) in COVID-19 are related to both acute stage and long-COVID. We explored levels of brain injury biomarkers (NfL and GFAP) and myeloid activation marker (sCD163) and their implications on the CNS. **Materials and Methods:** In hospitalized COVID-19 patients plasma samples were collected at two time points: on hospital admission (baseline) and three months after hospital discharge (Tpost). Patients were stratified according to COVID-19 severity based on acute respiratory distress syndrome (ARDS) onset (severe and non-severe groups). A further stratification according to the presence of NS (with and without groups) at baseline (requiring a puncture lumbar for diagnostic purposes) and according to NS self-referred at Tpost was performed. Finally, cerebrospinal fluid (CSF) samples were collected from patients with NS present at baseline. **Results:** We enrolled 144 COVID-19 patients (62 female/82 male; median age [interquartile range, IQR]): 64 [55–77]) and 53 heathy donors (HD, 30 female/23 male; median age [IQR]: 64 [59–69]). At baseline, higher plasma levels of NfL, GFAP and sCD163 in COVID-19 patients compared to HD were observed (*p* < 0.0001, *p* < 0.0001 and *p* < 0.0001, respectively), especially in those with severe COVID-19 (*p* < 0.0001, *p* < 0.0001 and *p* < 0.0001, respectively). Patients with NS showed higher plasma levels of NfL, GFAP and sCD163 compared to those without (*p* = 0.0023, *p* < 0.0001 and 0.0370, respectively). At baseline, in COVID-19 patients with NS, positive correlations between CSF levels of sCD163 and CSF levels of NfL (ρ = 0.7536, *p* = 0.0017) and GFAP were observed (ρ = 0.7036, *p* = 0.0045). At Tpost, the longitudinal evaluation performed on 77 COVID-19 patients showed a significant reduction in plasma levels of NfL, GFAP and sCD163 compared to baseline (*p* < 0.0001, *p* < 0.0001 and *p* = 0.0413, respectively). Finally, at Tpost, in the severe group, higher plasma levels of sCD163 in patients with NS compared to those without were reported (*p* < 0.0001). **Conclusions:** High plasma levels of NfL, GFAP and sCD163 could be due to a proinflammatory systemic and brain response involving microglial activation and subsequent CNS damage. Our data highlight the association between myeloid activation and CNS perturbations.

## 1. Introduction

Coronavirus disease 2019 (COVID-19) is a multisystem viral sepsis syndrome that can affect different organ systems with symptoms ranging from mild to life threatening [1]. Neurologic complications are commonly described and may occur as direct or indirect consequences of the viral infection, the treatment, the systemic inflammation due to immune activation or hypoxia or, in some cases, may be incidental associations [2,3,4,5]. However, the severe acute respiratory syndrome coronavirus 2 (SARS-CoV-2) is known to have neuroinvasive potential [6,7].

The accumulating data describing various neurological manifestations in COVID-19 are not only related to the acute phase [3,8,9,10] but are often part of a syndrome known as post-acute sequelae of COVID-19 (PASC) or long-COVID, with symptoms occurring and/or persisting for weeks or months after the initial infection [8,11,12,13,14]. To date, whether the neurological manifestations are also accompanied by increased biomarkers of neuronal and astrocytic damage is still being investigated.

Brain injury biomarkers of neurological diseases have been investigated in the context of COVID-19. Indeed, during the acute stage of COVID-19, both cerebrospinal fluid (CSF) [2,15,16] and plasma studies [2,17,18,19,20] showed higher levels of neurofilament light chain (NfL) in patients with neurological symptoms (NS) which are correlated with disease activity, thus further supporting the occurrence of concomitant acute axonal injury. NfL is a subunit of neurofilaments, which are cylindrical proteins exclusively located in the neuronal axons, that can be measured in both CSF and plasma samples as a biomarker of neuronal injury [21,22]. Likewise, in the acute phase of COVID-19, it has been shown that increased levels of glial fibrillary acidic protein (GFAP) correlate with COVID-19 severity [18,23,24,25]. GFAP is an intermediate filament highly expressed in astrocytes regulating their morphology and function in the CNS [26,27]. Plasma GFAP levels are very low in healthy individuals, but increased GFAP levels due to astrocyte disintegration are known to indicate astrocyte damage [26,28]. Therefore, GFAP is increasingly used as a plasma biomarker of astroglial activation/injury [21,28].

Alteration of the blood–brain barrier (BBB) integrity has an important role in neurological diseases, including brain infections [29]. Matrix metalloproteinases (MMPs), a family of enzymes that proteolytically degrade various components of the extracellular matrix (ECM), can have a detrimental effect, contributing to perturbation of the BBB integrity and neuroinflammation [30]. Within the ECM, the tissue inhibitors of MMPs (TIMPs) inhibit and regulate MMP proteolytic activity. Among MMPs, MMP-9 is the most prominent in promoting BBB disruption associated with CNS damage and inflammation [31]. In the acute stage of COVID-19, CSF levels of MMP-9 were found to increase and related to different inflammatory cytokines [32] as well as to COVID-19 severity [2].

Also, neurological diseases are associated with inflammatory conditions and several differences in the levels of biomarkers in neurological diseases compared to normal conditions were reported [33]. Among these biomarkers, soluble CD163 (sCD163) levels increase in both CSF and plasma samples, related to the activation of macrophage/microglia and inflammation levels in neurological diseases, supporting its role also as a cognitive impairment biomarker [34,35,36,37]. sCD163 is a soluble form of CD163, a receptor for haptoglobin–hemoglobin complexes and a monocyte/macrophage-specific membrane protein. CD163 is found on macrophages in the CNS, including perivascular macrophages and microglia [38] and its soluble form can be found in both CSF and plasma [39] after monocyte/macrophage activation [40,41]. In the CNS, sCD163 is probably shed by macrophages and microglia, triggered by complex immune-modulating mechanisms in the microenvironment [34]. Although plasma levels of sCD163 were deeply investigated in the acute stage of COVID-19 and correlate to COVID-19 severity [42,43,44], data on CSF levels of sCD163 are lacking.

In the present study, in a cohort of patients with different COVID-19 severity, we longitudinally investigated the levels of brain injury (NfL and GFAP), myeloid activation marker (sCD163) and in turn, implications for the CNS during the acute stage of the disease and three months after hospital discharge. Moreover, in those patients requiring puncture lumbar (PL) for diagnostic purposes, the impairment of blood–brain barrier (BBB) (MMP-9 and TIMP-1) was also investigated.

## 2. Materials and Methods

### 2.1. Study Design

A single-center retrospective study involving hospitalized COVID-19 patients from March 2020 to March 2021 was performed. Specifically, at the Department of Public Health and Infectious Diseases, Policlinico Umberto I, Sapienza, University of Rome, (baseline) adult patients (≥18 years old) were enrolled upon hospital admission. As previously described [45,46], COVID-19-related pneumonia was diagnosed by high-resolution chest computed tomography (CT) scan associated with SARS-CoV-2 RNA detection from a nasopharyngeal swab through a commercial reverse transcription-polymerase chain reaction (RT-PCR) kit, following manufacturer’s instructions (RealStar^®^ SARS-CoV-2 Altona Diagnostic, Hamburg, Germany).

Three months after hospital discharge (Tpost), COVID-19 patients were followed up with at the post-COVID clinic of Policlinico Umberto I, Sapienza, University of Rome. During the post-COVID visit, each subject underwent a detailed interview with an infectious disease physician who asked questions related to post-COVID symptoms. Specifically, patients were queried regarding the presence of post-COVID symptoms, including NS such as trouble concentrating or with memory, headache, trouble with taste or smell.

At first, according to COVID-19 severity at the acute stage of the disease (based on acute respiratory distress syndrome [ARDS] onset), patients were stratified into two groups: severe and non-severe. Then, these two groups were further classified into four subgroups based on the maximum oxygen supply/ventilation support required during hospitalization. Specifically, the severe group was stratified into invasive mechanical ventilation via orotracheal intubation (IOT) and noninvasive ventilation (NIV) subgroups while the non-severe group was stratified into Venturi mask for oxygen (VMK) and room air (AA) subgroups.

Next, an additional stratification was performed based on NS presence at both time points into groups with and without NS. Specifically, at baseline NS presence was defined according to the necessity of PL for diagnostic purposes, while at Tpost, NS was self-referred by the patients during the post-COVID visit after being asked by an infectious disease physician.

Finally, as a control group, healthy donors (HD) matched for age and sex, with a negative nasopharyngeal swab for SARS-CoV-2 RNA detection, undetectable anti-SARS-CoV-2-specific IgG and without any symptoms, were enrolled.

### 2.2. Data and Sample Collection

An ad hoc electronic database was created to collect demographic data, comorbidities, laboratory results, oxygen support and type of ventilation. Two time points were considered: hospital admission (baseline) and post-COVID-19 visit at three months after discharge (Tpost). All blood tests were performed in the hospital’s central laboratory following standard procedures.

### 2.3. Microfluidic Next Generation Enzyme-Linked Immunosorbent Assay (ELISA)

During routine clinical testing, CSF samples were collected in sterile tubes without anticoagulant while peripheral whole blood was collected in heparin-coated BD Vacutainer Blood Collection tubes (BD Biosciences, Franklin Lakes, NJ, USA). All samples were collected between 8 and 10 a.m. As previously described [2,47], in collected samples, the evaluation of NfL, GFAP, sCD163, MMP-9 and TIMP-1 levels was assessed using the Simple Plex^TM^ Ella Assay (ProteinSimple, San Jose, CA, USA) on Ella^TM^ microfluidic system (Bio-Techne, Minneapolis, MN, USA) according to the manufacturer’s instructions. Ella^TM^ was calibrated using the in-cartridge factory standard curve. The limits of detection of NfL, GFAP, sCD163, MMP-9, and TIMP-1 were 1.09 pg/mL, 14.4 pg/mL, 318 pg/mL, 10.5 pg/mL, and 0.34 pg/mL, respectively. The limits of detection were calculated by adding three standard deviations to the mean background signal determined from multiple runs.

### 2.4. Statistical Analysis

Analysis was performed with Prism 9 (GraphPad, Boston, MA, USA). A probability value < 0.05 was considered as statistically significant and NfL, GFAP and sCD163 were examined as continuous variables. Patient characteristics were compared using Student’s *t*-test or chi-square for continuous and categorical variables, respectively. Continuous variables were expressed as the median and interquartile range (IQR) with the assumption of a normal distribution. Categorical variables were expressed as counts and percentages. Groups were then compared using Student’s *t*-test or the Mann–Whitney U-test, as appropriate. The nonparametric Kruskal–Wallis test with Dunn’s post-test was used for comparing medians of groups and subgroups with HD as well as for comparing medians of IOT, NIV and VMK with AA subgroup. The nonparametric Wilcoxon test was used for the longitudinal evaluation comparing baseline and Tpost. Correlations were performed using Spearman rank correlation analysis.

## 3. Results

### 3.1. Study Population

One hundred and forty-four hospitalized COVID-19 patients and 53 HD were enrolled (Table 1). According to a chest CT scan, all COVID-19 patients had interstitial pneumonia and at least one confirmed positive molecular test for SARS-CoV-2 RNA detection using a nasopharyngeal swab. During hospitalization, 51.4% (74/144) developed ARDS requiring oxygen supply/ventilation and 18.8% (27/144) of COVID-19 patients died.

On hospital admission, the main clinical condition was SARS-CoV-2-related pneumonia, accounting for 89.6% (129/144) of the patients while for the remaining 15 COVID-19 patients the hospitalization was also due to severe NS and a PL for clinical purposes was performed.

At first, patients were stratified into severe (74/144) and non-severe (70/144) groups, according to ARDS onset during hospitalization. Then, these groups were further categorized into four subgroups based on the maximum oxygen supply/ventilation support required during hospitalization: IOT (9/144), NIV (55/144), VMK (31/144) and AA groups (39/144) (Table 1).

As reported in Table 1, patients in the IOT and NIV subgroups were older compared to patients in the AA subgroup (*p* = 0.0115 and *p* = 0.002, respectively). Among the four subgroups, no differences in the percentages of COVID-19 patients with at least one comorbidity were observed (Table 1). On hospital admission, a higher percentage of COVID-19 patients with shortness of breath in was observed in the IOT subgroup compared to the other subgroups (IOT: 47.4%, NIV: 36.4%, VMK: 19.4% and AA: 17.4%; *p* = 0.0412). Conversely, a higher percentage of COVID-19 patients with ageusia and anosmia was found in the AA subgroup compared to the other subgroups (IOT: 1.1%, NIV: 3.6%, VMK: 19.4% and AA: 20.5%; *p* = 0.0315) (Table 1). No differences in the percentages of COVID-19 patients with fever, cough, myalgia or arthralgia, diarrhea, sputum production and NS were found among the four subgroups (Table 1).

### 3.2. Plasma Biomarkers of Brain Injury on Hospital Admission

NfL and sCD163 were detectable in all collected plasma samples. However, GFAP was detectable only in 58.3% (88/144) of collected plasma samples.

On hospital admission, higher plasma levels of NfL, GFAP and sCD163 were observed in COVID-19 patients compared to HD (median values and [IQR] for NfL: 29 [15–66] and 7 [4–11], *p* < 0.00001; median values and [IQR] for GFAP: 2 [0–10] and 0 [0–0], *p* < 0.0001; median values and [IQR] for sCD163: 1458 [1119–2256] and 952 [590–1303], *p* < 0.0001). After stratifying COVID-19 patients according to COVID-19 severity and the maximum oxygen supply/ventilation support required during hospitalization, higher plasma levels of NfL, GFAP, and sCD163 in the severe group compared to the non-severe one were observed (median values and [IQR] for NfL: 45 [23–98] and 20 [13–33], respectively; *p* < 0.0001; median values and [IQR] for GFAP: 7 [3–16] and 0 [0–0], respectively; *p* < 0.0001; median values and [IQR] for sCD163: 1887 [1291–2494] and 1241 [1043–1707], respectively; *p* < 0.0001) (Figure 1B) as well as in IOT and NIV subgroups compared to the AA one (NfL: *p* = 0.0003 and *p* = 0.0011, respectively; GFAP: *p* < 0.0001 and *p* < 0.0001, respectively; sCD163: *p* = 0.0162 and *p* = 0.0076, respectively) (Figure 1B, Table 2). No differences between the VMK and AA subgroups were found (Figure 1B, Table 2).

Compared to HD, higher plasma levels of NfL and sCD163 in both severe and non-severe groups were observed (NfL: *p* < 0.0001 and *p* < 0.0001; sCD163: *p* < 0.0001 and *p* = 0.0021, respectively) (Figure 1B). Conversely, higher plasma levels of GFAP only in the severe group compared to HD were observed (*p* < 0.0001) (Figure 1B). Moreover, higher plasma levels of NfL and sCD163 in each subgroup compared to HD were observed (IOT: *p* < 0.0001 and *p* < 0.0001, respectively; NIV: *p* < 0.0001 and *p* < 0.0001, respectively; VMK: *p* < 0.0001 and *p* = 0.0241; AA: *p* < 0.0001 and *p* = 0.0451, respectively) (Figure 1B, Table 2). Conversely, compared to HD, higher plasma levels of GFAP were observed only in the IOT and NIV subgroups (*p* < 0.0001 and *p* < 0.0001, respectively) (Figure 1B, Table 2). Otherwise, no significant differences between the VMK and AA subgroups compared to HD were observed (Figure 1B, Table 2).

When stratifying patients according to NS presence on hospital admission, higher plasma levels of NfL, GFAP and sCD163 were observed in patients with NS compared to those without (NfL: 84 [30–97] and 29 [14–62], respectively, *p* = 0.0023; GFAP: 13 [11–16] and 1 [0–6], respectively, *p* < 0.0001; sCD163: 2009 [1453–2524], respectively, *p* = 0.0370). The stratification of both severe and non-severe groups was performed based on the presence of NS on hospital admission (Figure 1C, Table 3). Higher plasma levels of NfL and GFAP in patients with NS compared to patients without NS were observed in both groups (severe group: *p* = 0.0575 and *p* = 0.0051, respectively; non-severe-group: *p* = 0.0427 and *p* < 0.0001, respectively) (Figure 1C, Table 3). Otherwise, no statistical differences in plasma levels of sCD163 were found (Figure 1C, Table 3).

Finally, among the 15 COVID-19 patients with severe NS on hospital admission requiring PL for diagnostic purposes, an investigation of neuronal and astrocyte damage (NfL and GFAP), myeloid activation (sCD163) and BBB alteration biomarkers (MMP-9 and TIMP-1) on CSF samples was performed (Table 4). All biomarkers were detectable in all CSF samples (Table 4).

A positive correlation between CSF and plasma NfL levels was found (ρ = 0.8357, *p* = 0.0002) as well as between CSF levels of NfL and sCD163 (ρ = 0.7536, *p* = 0.0017) (Table 4). Similarly, positive correlations between CSF and plasma GFAP levels were found (ρ = 0.8739, *p* < 0.0001) and between CSF levels of GFAP and sCD163 (ρ = 0.7036, *p* = 0.0045) (Table 4). Moreover, a positive correlation between CSF levels of GFAP and MMP-9 was observed (ρ = 0.5870, *p* = 0.0242) (Table 4). Finally, positive correlations between CSF levels of sCD163 and MMP-9 (ρ = 0.5786, *p* = 0.0263) and between CSF levels of sCD163 and TIMP-1 were found (ρ = 0.6786, *p* = 0.0068) (Table 4).

### 3.3. Post-COVID Symptoms

After three months from hospital discharge (Tpost), all the 121 living COVID-19 patients were invited via telephone to the post-COVID clinic (Appendix A). Among them, 44 COVID-19 patients declined while 77 patients were evaluated at post-COVID clinic. The demographic and clinical characteristics of the 77 COVID-19 patients are shown in Appendix A.

As reported in Figure 2, during the post-COVID visit, 63/77 patients self-referred at least one symptom. Specifically, below general symptoms such as fatigue, malaise and asthenia mentioned by 28/77 patients, the most prevalent symptoms involved the respiratory system, such as a cough and shortness of breath in 22/77 patients, and the nervous system, such as cognitive impairment, headache and loss of smell and taste in 22/77 patients. Immediately following in the list, 21/77 patients mentioned mental health symptoms, such as anxiety, depression, and sleep problems, and 12/77 patients mentioned cardiovascular symptoms, such as arrhythmia and palpitation. Finally, 10/77 patients mentioned skin disorders, such as rash and hair loss, and 4/77 reported musculoskeletal symptoms such as joint pain and muscle weakness (Figure 2).

### 3.4. Plasma Biomarkers of Brain Injury in Long COVID

Overall, at Tpost, a significant reduction was observed in plasma levels of NfL, GFAP and sCD163 compared to the baseline (median values and [IQR] for NfL: 21 [12–35] and 17 [10–23], respectively, *p* < 0.0001; median values and [IQR] for GFAP: 0 [0–3] and 0 [0–1], respectively, *p* < 0.0001; median values and [IQR] for sCD163: 1531 [1123–2031] and 1121 [754–1708], respectively, *p* = 0.0413) (Figure 3A). However, compared to HD at Tpost, plasma levels of NfL, GFAP and sCD163 were almost higher (*p* < 0.0001, *p* = 0.0045, and *p* = 0.0418) (Figure 3A). The same results were also observed excluding COVID-19 patients with renal and pulmonary comorbidities (Appendix A).

According to COVID-19 severity at Tpost, in both severe and non-severe groups a significant reduction in plasma levels of NfL and sCD163 was observed compared to baseline (NfL: *p* = 0.0008 and *p* = 0.0003, respectively; sCD163: *p* = 0.0240 and *p* = 0.0014, respectively) (Table 5). Otherwise, at Tpost, plasma levels of GFAP were only significantly reduced in the severe group while no differences were observed in the non-severe group (Table 5). At Tpost, compared to HD, higher plasma levels of NfL in both severe and non-severe groups were observed (*p* < 0.0001 and *p* = 0.0046, respectively) while plasma levels of GFAP and sCD163 were higher only in the severe group (*p* = 0.0040 and *p* = 0.0003, respectively) (Figure 3B, Table 5).

As reported in Figure 3B, a further longitudinal evaluation of plasma levels of NfL, GFAP and sCD163 was performed according to the maximum oxygen supply/ventilation support required during hospitalization. At baseline, plasma levels of NfL and sCD163 were higher in each subgroup compared to HD, as reported in the previous paragraph on the whole study population (IOT: *p* = 0.0005, and *p* = 0.0093, respectively; NIV: *p* < 0.0001, and *p* < 0.0001, respectively; VMK: *p* = 0.001 and *p* = 0.0119, respectively; AA: 0.0007 and *p* = 0.0021, respectively) (Figure 3B). Moreover, at baseline higher plasma levels of GFAP was found only in the IOT and NIV subgroups compared to HD (*p* < 0.0001 and *p* < 0.0001, respectively) (Figure 3B).

At Tpost, no statistical differences in the IOT subgroup compared to baseline were observed (Table 5). Conversely, in the NIV subgroup, a significant reduction in plasma levels of NfL, GFAP and sCD163 was observed at Tpost compared to baseline (*p* = 0.0006, *p* < 0.0001, and *p* = 0.0046, respectively) (Figure 3B, Table 5). Similarly, in the VMK and AA subgroups a significant reduction in plasma levels of NfL and sCD163 at Tpost compared to baseline was found (VMK: *p* = 0.0258 and *p* = 0.0108, respectively; AA: *p* = 0.0035 and *p* = 0.0457, respectively) (Figure 3B, Table 5). On the other hand, no differences in plasma levels of GFAP in the VMK and AA subgroups was observed (Figure 3B, Table 5).

Compared to HD, at Tpost, plasma levels of NfL were higher in each subgroup (IOT: *p* = 0.0005, NIV: *p* < 0.0001, VMK: *p* = 0.0011 and AA: *p* = 0.0007) while higher plasma levels of GFAP were observed only in the IOT and NIV subgroups (*p* = 0.0399 and *p* < 0.0001, respectively) (Figure 3B, Table 5) and higher plasma levels of sCD163 were found only in the IOT subgroup (*p* = 0.0010) (Figure 3B, Table 5).

Among the 15 COVID-19 patients with NS on hospital admission, only 5 accepted the post-COVID visit (Appendix A). As reported in Appendix A, in the longitudinal evaluation both groups (with and without NS on hospital admission) showed a reduction in plasma levels of NfL and GFAP over time. On the other hand, a decrease in plasma levels of sCD163 was observed only in patients without NS on hospital admission while no difference in plasma levels of sCD163 was found among patients with NS (Appendix A).

Finally, stratifying the severe and non-severe groups according to the presence of at least one self-referred NS at Tpost, a higher plasma level of sCD163 in patients with NS compared to patients without NS was observed only in the severe group (*p* < 0.0001) (Figure 3C, Table 6). Intriguingly, no differences in plasma levels of NfL and GFAP were found by comparing patients with and without NS in both severe and non-severe groups (Figure 3C, Table 6).

## 4. Discussion

In this study, to investigate the influence of hypoxia on the levels of neuronal and astrocytic injury and myeloid activation biomarkers, we stratified the study population according to both COVID-19 severity and maximum oxygen supply/ventilation support required during hospitalization. A further stratification was performed based on the presence of NS at both time points. Finally, an investigation of neuronal (NfL) and astrocyte damage (GFAP), myeloid activation (sCD163) and BBB permeability (MMP-9 and TIMP-1) was performed on the CSF samples of hospitalized COVID-19 patients with severe NS on hospital admission.

Our key findings are that: (1) on hospital admission, plasma levels of NfL, GFAP and sCD163 were significantly higher in COVID-19 patients compared to HD, especially in those who developed a severe form of the disease requiring ventilation support during hospitalization; (2) plasma levels of NfL and GFAP were significantly higher in patients with NS at the acute stage of the disease, compared to those without NS but with equal COVID-19 severity; (3) in COVID-19 patients with severe NS on hospital admission, positive correlations between CSF levels of sCD163 and CSF levels of NfL, GFAP and MMP-9 were observed; (4) plasma levels of NfL, GFAP and sCD163 were significantly reduced three months after hospital discharge, although levels were still higher compared to HD, especially in those with a severe form in the acute stage of COVID-19; (5) three months after hospital discharge, NS were self-reported by patients, and among them, patients with a severe form in the acute stage of COVID-19 showed higher plasma levels of sCD163 compared to those without NS. Conversely, no differences in plasma levels of NfL and GFAP were observed.

Consistent with previously published data, our results confirmed that plasma levels of CNS damage biomarkers were higher in hospitalized patients with severe COVID-19 [18,20,48,49,50,51]. According to the existing literature, in the acute stage of COVID-19 neurological involvement is accompanied by CNS damage, although the direct infection of brain parenchyma by SARS-CoV-2 remains a debated issue [2,52]. In addition to possible direct viral invasion, several mechanisms likely contribute to CNS involvement in COVID-19, including the indirect effects of systemic inflammation due to immune activation or hypoxia as a hematogenous pathway [3,4,5]. To date, there is increasing evidence that neuronal injury is mediated primarily via hypoxia [2] and hyperinflammation [5]. Overall, our data confirmed that hypoxemia and systemic inflammation could be possible causes of CNS injury.

As a marker of inflammation, sCD163 is elevated in a range of diseases [34,43,53,54] and it was largely investigated in COVID-19 pneumonia [42,43,44,55]. As previously reported, a potential use to assess the risk of disease progression has been proposed, because an increase in plasma levels of sCD163 was observed on hospital admission in COVID-19 patients, especially in those who developed ARDS, as well as its correlation with typical inflammatory markers of COVID-19 pneumonia [43,44]. To the best of our knowledge, there is no information on the assessment of sCD163 in the CSF of COVID-19 patients nor on its correlation with NS at both the acute stage and post-COVID. However, our data seem to confirm the potential role of sCD163 as a CNS impairment biomarker. sCD163 is known to be upregulated during the pro-inflammatory response, and the release of MMPs is known to contribute to this orchestration [56]. Thus, despite the small sample size, we observed a positive correlation between the CSF levels of sCD163 and MMP-9, underlining that increased CSF levels of sCD163 and MMP-9 might also contribute to the infiltration of monocytes to the CSF in COVID-19 patients. In addition, the positive correlations between CSF levels of sCD163 and CSF levels of both NfL and GFAP underline the ongoing inflammation, as postulated in other neurological conditions [34]. Indeed, sCD163 levels in the CSF have been shown to be elevated in patients with multiple sclerosis as well as in combination antiretroviral therapy-treated people living with HIV with a mild neurocognitive disorder. Plasma sCD163 levels were elevated compared to those who are cognitively normal or who have asymptomatic neurocognitive impairment [57]. Also, in neurodegenerative disorders, such as Parkinson’s disease, sCD163 is a potential cognition-related biomarker underlighting a role of monocytes in both peripheral and brain immune responses [35].

Finally, positive correlations between the CSF and plasma levels of NfL and GFAP in COVID-19 patients with severe NS on hospital admission, validate their plasma assessment as a less-invasive biomarker for diagnosis, prognosis and monitoring of CNS damage associated with COVID-19. Moreover, activated astrocytes upregulate GFAP expression, produce fine processes, and exhibit hypertrophic morphology [58]. In addition to the role of active astrocytes in the formation of the physical BBB, aberrant ECM proteins at the lesion site also inhibit reparative precursor cell migration during recovery [59,60]. MMP-9 plays a key role in ECM remodeling during brain injury due to its capacity to proteolytically degrade ECM [61]. Our data underline the detrimental effects of MMP-9, including BBB disruption and inflammation, in the acute phase of COVID-19 associated with astrocyte damage.

To clarify the role of NfL, GFAP and sCD163 as a sign for CNS damage presenting with NS, we stratified hospitalized COVID-19 patients according to the severe presence of NS during the acute stage of COVID-19 and self-referred NS three months after hospital discharge. During the acute stage of the disease, patients with severe NS (requiring puncture lumbar for diagnostic purposes) showed higher plasma levels of NfL, GFAP and sCD163 compared to patients without severe NS. However, three months after hospital discharge, no differences in plasma levels of both NfL and GFAP between patients with and without self-referred NS were found. However, among severe COVID-19 patients, the high plasma levels of NfL and GFAP in patients without severe NS suggest the presence of subclinical central nervous system involvement. Conversely, among patients with severe COVID-19 at the acute stage, high plasma levels of sCD163 seem to be associated with self-referred NS. The elevated levels of sCD163 could be a mechanism of tissue homeostasis and repair and thus sCD163 could be a marker of immune modulatory functions regarding not only degeneration and anti-inflammation but also tissue repair [62,63,64].

Finally, the longitudinal evaluation showed that despite the significant reduction in plasma levels of CNS injury biomarkers, three months after hospital discharge, COVID-19 patients still have persistently higher levels compared to HD. These data are in line with previous reports [2,65] although Kanberg et al. [50] reported a complete normalization of CNS injury biomarkers after six months post-infection.

Our single-center study has limitations such as the small sample size of patients with severe NS at the acute stage of COVID-19 and the evaluation of self-referred NS three months after hospital discharge, which was not confirmed by standardized cognitive tests but recorded by an infectious disease physician. Finally, only half of the enrolled patients were evaluated at the post-COVID clinic and among them just 5 out 15 patients had severe NS at the acute stage of COVID-19.

Overall, plasma biomarkers of brain injury, NFL and GFAP, as well as monocyte/macrophage activation markers have been found to be increased in a severity-dependent manner in hospitalized COVID-19 patients. High plasma and CSF levels of NfL and GFAP in COVID-19 could be due to a proinflammatory systemic and brain response that involves microglial activation and subsequent neuronal damage. Our data further highlight the association between myeloid activation and CNS perturbations.

The recognition and diagnosis of these neurologic complications at both acute-stage and post-COVID are challenging, particularly in the context of overstrained medical systems, where an under-recognition or delays in diagnosis may contribute to poor outcomes [12]. However, further studies are required to clarify the nature of CNS injury and evaluate the usefulness of these biomarkers in COVID-19 patients.

## Figures and Tables

**Figure 1 cells-12-02270-f001:**
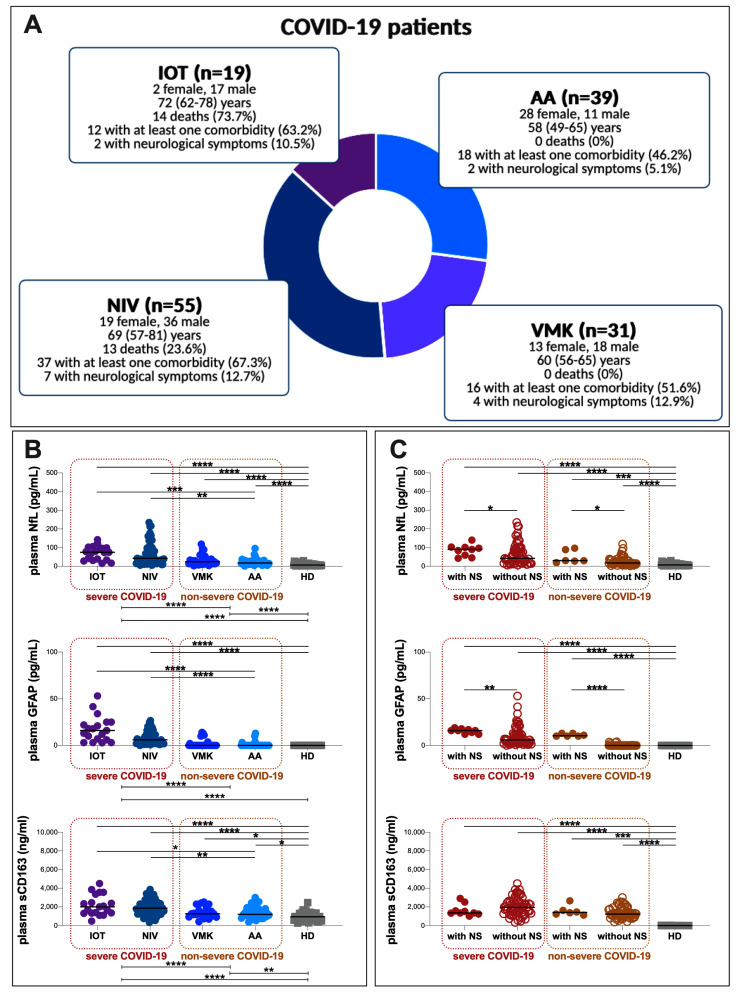
(**A**) Schematic representation of study population with principal features. (**B**) Evaluation of NfL, GFAP and sCD163 plasma levels in the study population according COVID-19 severity and maximum oxygen supply/ventilation support required during hospitalization and (**C**) according to severe NS on hospital admission. IOT: invasive mechanical ventilation via orotracheal intubation, NIV: noninvasive venous ventilation, VMK: Venturi mask for oxygen, AA: room air, HD: healthy donors, *n*: number, NfL: neurofilament light chain, GFAP: glial fibrillary acidic protein, sCD163: soluble CD163. *: 0.05 < *p* < 0.01; **: 0.01 < *p* < 0.001; ***: 0.001 < *p* < 0.0001; ****: *p* > 0.0001.

**Figure 2 cells-12-02270-f002:**
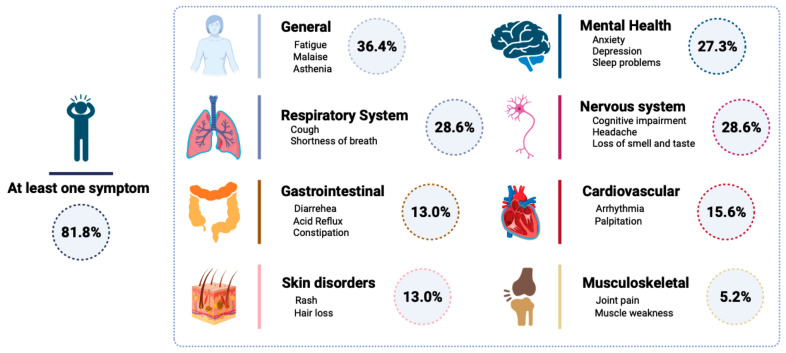
Schematic representation of self-referred post-COVID symptoms.

**Figure 3 cells-12-02270-f003:**
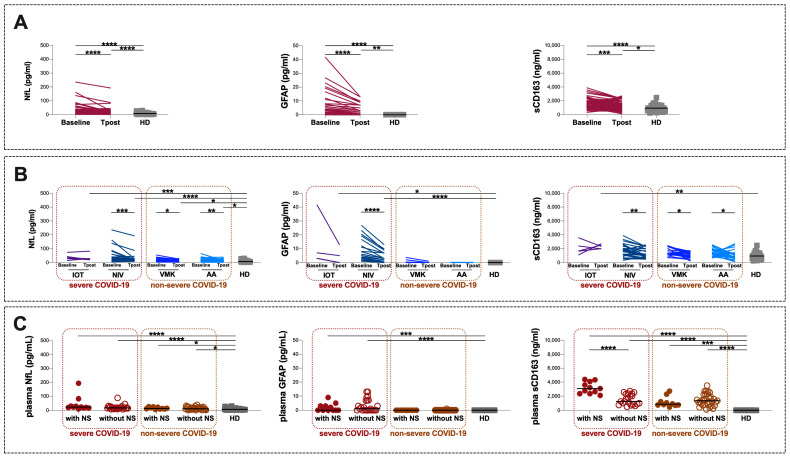
(**A**) Longitudinal evaluation of NfL, GFAP and sCD163 plasma levels in the study population (**B**) according to the maximum oxygen supply/ventilation support required during hospitalization and (**C**) according to both severity of COVID-19 at acute stage and self-referred NS during post-COVID visit. IOT: invasive mechanical ventilation via orotracheal intubation, NIV: noninvasive venous ventilation, VMK: Venturi mask for oxygen, AA: room air, HD: healthy donors, *n*: number, NfL: neurofilament light chain, GFAP: glial fibrillary acidic protein, sCD163: soluble CD163. *: 0.05 < *p* < 0.01; **: 0.01 < *p* < 0.001; ***: 0.001 < *p* < 0.0001; ****: *p* > 0.0001.

**Table 1 cells-12-02270-t001:** Study population.

	Severe (*n* = 74)		Non-Severe (*n* = 70)		
	IOT (*n* = 19)	NIV (*n* = 55)	VMK (*n* = 31)	AA (*n* = 39)	HD (*n* = 53)
**Female, *n* (%)**	2 (10.5)	19 (34.5)	13 (41.9)	28 (71.8)	30 (56.6)
**Age, median (IQR) (years)**	72 (62–78)	71 (58–82)	60 (56–70)	58 (49–65)	64 (59–69)
**Deaths, *n* (%)**	14 (73.7)	13 (26.6)	0 (0)	0 (0)	-
**Comorbidity**					
Any, *n* (%)	12 (63.2)	37 (67.3)	16 (51.6)	18 (46.2)	-
Hypertension, *n* (%)	8 (42.1)	20 (36.4)	8 (25.8)	14 (35.9)	-
Cardiovascular, *n* (%)	7 (36.8)	11 (20.0)	4 (12.9)	4 (10.3)	-
Diabetes, *n* (%)	5 (26.3)	9 (16.4)	3 (9.7)	3 (7.7)	-
Pulmonary, *n* (%)	3 (15.8)	8 (14.5)	3 (9.7)	4 (10.3)	-
Cancer, *n* (%)	1 (5.3)	10 (18.2)	2 (6.5)	1 (2.6)	-
Renal, *n* (%)	0 (0)	5 (9.1)	0 (0)	1 (2.6)	-
**Symptoms**					
Fever, *n* (%)	14 (73.7)	38 (69.1)	22 (71.0)	27 (69.2)	-
Cough, *n* (%)	7 (36.8)	23 (41.8)	10 (32.3)	15 (38.5)	-
Shortness of breath, *n* (%)	9 (47.4)	20 (36.4)	6 (19.4)	7 (17.9)	-
Myalgia or arthralgia, *n* (%)	5 (26.3)	9 (16.4)	11 (35.5)	10 (25.6)	-
Diarrhea, *n* (%)	9 (47.4)	6 (10.9)	6 (19.4)	3 (7.7)	-
Anosmia and ageusia, *n* (%)	1 (5.3)	2 (3.6)	6 (19.4)	8 (20.5)	-
Sputum production, *n* (%)	0 (0)	2 (3.6)	0 (0)	1 (2.6)	-
Neurological, *n* (%)	2 (10.5)	7 (12.7)	4 (12.9)	2 (5.1)	-

*n*: number, IQR: interquartile range, IOT: invasive mechanical ventilation via orotracheal intubation, NIV: noninvasive venous ventilation, VMK: Venturi mask for oxygen, AA: room air, HD: healthy donors.

**Table 2 cells-12-02270-t002:** Next-generation ELISA data according to COVID-19 severity and maximum oxygen supply/ventilation support required during hospitalization.

	Severe (*n* = 74)		Non-Severe (*n* = 70)		
	IOT (*n* = 19)	NIV (*n* = 55)	VMK (*n* = 31)	AA (*n* = 39)	HD (*n* = 53)
Plasma NfL (pg/mL)	75 (34–99)	43 (22–97)	25 (14–54)	18 (12–31)	7 (4–11)
Plasma GFAP (pg/mL)	16 (6–25)	6 (2–12)	0 (0–0)	0 (0–0)	0 (0–0)
Plasma sCD163 (ng/mL)	2011 (1336–3360)	1867 (1226–2486)	1268 (1023–1743)	1208 (946–1711)	952 (590–1303)

*n*: number, IOT: invasive mechanical ventilation via orotracheal intubation, NIV: noninvasive venous ventilation, VMK: Venturi mask for oxygen, AA: room air, HD: healthy donors, NfL: neurofilament light chain, GFAP: glial fibrillary acidic protein, sCD163: soluble CD163. Data are shown as median (interquartile range).

**Table 3 cells-12-02270-t003:** Next-generation ELISA data according to severe NS on hospital admission.

	Severe (*n* = 74)		Non-Severe (*n* = 70)		
	with NS (*n* = 9)	without NS (*n* = 65)	with NS (*n* = 6)	without NS (*n* = 64)	HD (*n* = 53)
Plasma NfL (pg/mL)	90 (52–102)	43 (22–97)	30 (25–91)	18 (12–33)	7 (4–11)
Plasma GFAP (pg/mL)	16 (13–17)	6 (2–12)	1 (10–13)	0 (0–0)	0 (0–0)
Plasma sCD163 (ng/mL)	1353 (1284–2050)	1944 (1278–2503)	1415 (1204–1879)	1234 (994–1716)	952 (590–1303)

*n*: number, NS: neurological symptoms; with: patients with severe NS on hospital admission, without: patients without severe NS on hospital admission, HD: healthy donors, NfL: neurofilament light chain, GFAP: glial fibrillary acidic protein, sCD163: soluble CD163. Data are shown as median (interquartile range).

**Table 4 cells-12-02270-t004:** Investigation of CSF and plasma samples of COVID-19 patients with severe NS on hospital admission.

					NfL (pg/mL)		GFAP (pg/mL)		sCD163 (ng/mL)		MMP-9 (pg/mL)	TIMP-1 (pg/mL)
Patient	Gender	Age	NS on Hospital Admission	Ventilation Support	CSF	Plasma	CSF	Plasma	CSF	Plasma	CSF	
**1**	male	67	confusion	VMK	330	30	650	10	23	1621	1219	1346
**2**	male	83	confusion, syncope	NIV	889	84	764	12	43	1576	9	49,004
**3**	female	70	headache, confusion	AA	622	27	662	10	38	1232	28	48,354
**4**	female	61	nystagmus, seizure, forced deviation of the left to the left	VMK	1591	89	708	13	73	2654	104	68,715
**5**	male	86	weakness, headache, gaze deviation to the right	NIV	7961	140	4336	16	285	1345	733	106,787
**6**	female	58	headache, confusion	AA	318	20	570	13	32	1120	71	134
**7**	male	36	headache, confusion	IOT	4998	90	80,754	18	245	1336	15,871	455,431
**8**	male	69	lower limb paresthesia	NIV	18,555	103	1004	12	365	1231	9118	697,953
**9**	female	67	headache, confusion	VMK	6661	97	995	11	216	1453	228	315,212
**10**	female	62	impaired bilateral vision and frontal headache	NIV	8497	60	3560	14	51	2910	4468	32,304
**11**	male	78	headache, confusion	IOT	1720	97	2150	16	86	1353	18	138,117
**12**	male	60	headache, confusion	VMK	266	30	567	10	44	1376	151	71,671
**13**	male	50	headache, confusion	NIV	456	45	4560	17	47	1523	526	98,811
**14**	male	43	confusion	NIV	432	43	5432	16	146	2524	548	100,108
**15**	male	54	headache, confusion	NIV	5998	101	50,754	19	342	1009	45,900	1346

NS: neurological symptoms, NfL: neurofilament light chain, GFAP: glial fibrillary acidic protein, sCD163: soluble CD163, IOT: invasive mechanical ventilation via orotracheal intubation, NIV: noninvasive venous ventilation, VMK: Venturi mask for oxygen, AA: room air, CSF: cerebrospinal fluid.

**Table 5 cells-12-02270-t005:** Next-generation ELISA data according to COVID-19 severity and maximum oxygen supply/ventilation support required during hospitalization among patients evaluated at post-COVID clinic.

	Severe (*n* = 34)				Non-Severe (*n* = 43)			
	Baseline		Tpost		Baseline		Tpost	
Plasma NfL (pg/mL)	28 (15–59)		19 (14–28)		15 (10–29)		13 (8–21)	
Plasma GFAP (pg/mL)	3 (2–9)		1 (0–6)		0 (0–0)		0 (0–0)	
Plasma sCD163 (ng/mL)	1746 (1203–2380)		1238 (912–2307)		1296 (1109–1925)		1062 (708–1436)	
	**IOT (*n* = 5)**		**NIV (*n* = 29)**		**VMK (*n* = 19)**		**AA (*n* = 24)**	
	**Baseline**	**Tpost**	**Baseline**	**Tpost**	**Baseline**	**Tpost**	**Baseline**	**Tpost**
Plasma NfL (pg/mL)	34 (22–58)	28 (21–54)	27 (15–60)	18 (13–23)	16 (10–30)	13 (10–19)	15 (11–29)	13 (7–21)
Plasma GFAP (pg/mL)	0 (0–3)	0 (0–9)	3 (2–10)	1 (0–5)	0 (0–0)	0 (0–0)	0 (0–0)	0 (0–0)
Plasma sCD163 (ng/mL)	3078 (2345–4084)	2345 (1996–2613)	1781 (1200–2360)	1065 (826–2009)	1273 (1176–2026)	1123 (708–1506)	1368 (1099–1898)	981 (702–1396)

*n*: number, IOT: invasive mechanical ventilation via orotracheal intubation, NIV: noninvasive venous ventilation, VMK: Venturi mask for oxygen, AA: room air, NfL: neurofilament light chain, GFAP: glial fibrillary acidic protein, sCD163: soluble CD163. Data are shown as median (interquartile range).

**Table 6 cells-12-02270-t006:** Next-generation ELISA data according to self-referred NS three months after hospital discharge.

	Severe (*n* = 34)		Non-Severe (*n* = 43)		
	with NS (*n* = 11)	without NS (*n* = 23)	with NS (*n* = 10)	without NS (*n* = 33)	HD (*n* = 53)
Plasma NfL (pg/mL)	23 (17–30)	18 (13–21)	14 (10–23)	11 (8–21)	7 (4–11)
Plasma GFAP (pg/mL)	0 (0–3)	1 (0–7)	0 (0–0)	0 (0–0)	0 (0–0)
Plasma sCD163 (ng/mL)	3078 (2345–4084)	1243 (854–2319)	856 (741–1473)	1351 (804–2086)	952 (590–1303)

*n*: number, NS: neurological symptoms; with: patients with self-referred NS during post-COVID visit three months after hospital discharge, without: patients without self-referred NS during post-COVID visit three months after hospital discharge, HD: heathy donors, NfL: neurofilament light chain, GFAP: glial fibrillary acidic protein, sCD163: soluble CD163. Data are shown as median (interquartile range).

## Data Availability

All data generated or analyzed during this study are included in this published article. The datasets used and/or analyzed during the current study are available from the corresponding author upon reasonable request.

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
