# Peer review of "Biomarkers of Neurological Damage: From Acute Stage to Post-Acute Sequelae of COVID-19"

_cells, 2023, doi:10.3390/cells12182270_

Round 1

Reviewer 1 Report

This interesting paper describes serum and CSF markers (NfL, GFAP, sCD163) of nervous system damage in patients with COVID19, stratifying patients by (pulmonary) severity and presence or absence of “nervous system symptoms”. The observations are intriguing and potentially important clues, but several key aspects require clarification.

1. First, labelling of Figures 1C and 3C would benefit from clarification.  I assume “severe’ and “non-severe” again refer to respiratory status and “with” and “without” refer to presence or absence of neurologic symptoms.

2. Assuming that is correct, it becomes important to provide more detail about the patients’ “neurologic symptoms”, particularly given the heterogeneity in marker concentrations demonstrated in these 2 figures, with markers minimally elevated or normal in most, substantially elevated in a few, the latter likely driving the mean values. In most patients SARS-CoV2 is not neuroinvasive. Neurologic symptoms in the acute setting most commonly are those seen in other patients with severe sepsis and/or respiratory failure and are the result of physiologic changes in brain function – which is both confounding and what makes these observations so potentially important.  The findings would be far more informative if patient groups were further subdivided into those 15 with prominent neurologic symptoms at admission, then subdividing “neurologic symptoms” into headache, toxic metabolic encephalopathy without imaging or other evidence of structural brain damage, hypoxic brain damage demonstrable on imaging, cerebrovascular events, cerebral hemorrhages or other apparent structural brain damage. Does stratification by neuropathology explain the heterogeneity in values, with abnormal elevations in those with structural neurologic damage?  If the 15 admitted with primarily nervous system difficulty are analyzed separately, do they account for the patients with elevations?

3. Interestingly, in Figure 1c it appears the patients with the most marked elevations in these markers were those without nervous system symptoms.  This requires clarification.

4. The relationship of PASC to nervous system damage is very unclear, making observations of this subgroup important. As a first step, it would be helpful to exclude those with other significant end organ damage - renal, pulmonary, etc. - which may be confounding. Then, it would again be helpful to stratify both by the presence or absence of true neurologic damage at onset (as above), and by specific putative nervous system symptoms in PASC patients. More specifically, of the patients with PASC, how many were among the 15 with primarily neurologic presentations initially?  Of those with primarily respiratory presentations, how many PASC patients had hypoxic or other structural neurologic damage at the outset?  And how did elevation of these markers at follow up correlate with specific “neurologic” symptoms – either at onset or at follow-up? Specifically, how many of those with elevated markers at follow up had focal neurologic residua? headache? cognitive symptoms? Anxiety? depression? Sleep problems?

5. Finally, it would be of interest to compare these findings to those in a control group with sepsis or respiratory failure, without structural nervous system damage

Overall excellent.  A few minor idiomatic issues

Reviewer 2 Report

This paper reports on a single study cohort study of clinical and biochemical neurological markers among their hospitalised COVID-19 patients. The authors found higher plasma levels of neuronal and astrocyte damage (NfL and GFAP) and myeloid activation (sCD163) in COVID-19 patients compared to healthy donors, especially in those with severe COVID-19. Patients with neurological symptoms (NS) showed higher plasma levels of NfL, GFAP and sCD163 34 compared to those without. When reviewed 3 months later, plasma levels of NfL, GFAP and sCD163 were significantly reduced compared to baseline. In addition, among 15 COVID-19 patients with severe NS on hospital admission requiring lumbar puncture for diagnostic purpose, biomarkers of neuronal and astrocyte damage (NfL and GFAP), myeloid activation (sCD163), and BBB alteration biomarkers (MMP-9 and TIMP-1) were detectable in all CSF samples.

There are a number of issues the authors may wish to address:
1. Abstract - Materials and Methods – line 29 - clarify how severe and non-severe groups were defined; also for NS in line 30. No mention made of the CSF assessment? Results – line 31 – to add demographics eg age and sex. Also to add the statistics. Line 36 – to mention that 77/144 were seen at Tpost.
2. Introduction – line 98 – biomarkers of impairment of blood-brain barrier (BBB) (MMP-9 and TIMP-1) were not done in all – to clarify in whom
3. Materials and Methods – 3.1 Study design – please state if this was a prospective cohort study or a retrospective cohort study. Line 118 – what is ‘venous’? Line 121 – how were NS determined? Patient volunteering the symptoms without being asked (stoic patients say nothing)? Patients asked if they had neuro symptoms? Questionnaire? Asked by a neurologist?
4. Table 1 – suggest use just Female and Deaths. Is age mean? median? Range interquartile? To show the data as % for ease of comparison. Are they statistically significantly differently?
5. Results – line 185 - ’otherwise’ - use ‘However’.  Line 239 - ‘PL’? Line 257 – only 77 of 144 were seen at Tpost – please explain who were the ones who came for review or were selected for review, and if they were different from those who did not (risk of selection bias)
6. Table 2 – are the ranges interquartile? Statistics?
7. Table 3 – are the ranges interquartile? Statistics? Please add HD data
8. Table 5 - are the ranges interquartile? Statistics? Please add HD data
9. Table 6 – line 319 - ‘self-referred’ – please explain, this info should be clearly explained in Methods too
10. Fig 1 A – use % where suitable, not numbers
11. Discussion – lines 333-344 – should be merged into the Introduction and not appear here. Limitations - line 420 – can also include single centre study, non-standardised evaluation of NS and not by neurologist, 50% drop out at follow-up

Reviewer 3 Report

In this manuscript, Maria et al used a large number of COVID-19 patient blood samples to screen for GFAP protein as a potential biomarker for patients infected with COVID-19 and for post-infection biomarkers. They also analyzed the post-recovery sequelae of COVID-19, which is of great significance for understanding the recovery process after COVID-19 and a topic of public interest. The study had a substantial amount of data, a long collection time, and yielded meaningful research results.

Regarding this article, I have only one question. As we know, the expression of GFAP is controlled by circadian rhythms. Therefore, with different sampling times for these patients, their clock phases would also vary, leading to varying levels of rhythmically regulated protein expression, including GFAP itself. If the study could incorporate time of sampling as a controlled factor, it would make the research even more interesting. Additionally, during the early stages of infection, the patients' biological clocks might be completely disrupted. Comparing the expression of clock proteins in the blood during the initial infection and after recovery would also be intriguing.

Round 2

Reviewer 1 Report

I thank the authors for their thoughtful responses to my queries.  I have just a few follow up suggestions, in light of their responses.

1. While it’s fair to describe these biomarkers as indicative of “neurological damage” it is misleading to refer to (subjective) "cognitive impairment, headache and loss of smell and taste” as “neurological symptoms”.  Headache rarely reflects neurologic damage, subjective cognitive impairment can include many non-neuropathologic disorders, and it appears the loss of taste and smell in COVID19 is typically due to sensory end organ changes and not neurologic ones.  Perhaps if symptoms were referred to as “patient-reported neurobehavioral changes” or something similar this might be less confusing.  It also deserves more prominent mention that data are based on just 5 of the 15 with neurobehavioral changes during acute disease, with the remainder declining (or lost to) follow up.  This heavily biases the data..

2. Figure 1 C.  Means/medians notwithstanding, inspection of the panels of patients with severe disease shows that the higher values were found preferentially in patients without “neurologic” symptoms.  Overall, the mean was pulled down by the large number of patients with lower values but clearly something was going on in a subset of these patients.  Some discussion of this would be informative.

Some modest editing will help

Reviewer 2 Report

This paper is a revised submission of a single centre retrospective cohort study of clinical and biochemical neurological markers among their hospitalised COVID-19 patients. The authors have addressed almost all my concerns, except these from before:

1. Previous line 121 (now 131-134) – it is still unclear to me how NS were determined - based on patients volunteering the symptoms without being asked (concern - stoic patients say nothing)? All patients directly being asked if they had neuro symptoms? Was a questionnaire used? Or only based on what was recorded in the case records? NS was determined to be present only when PL was deemed necessary? And am I right that at Tpost, it was only asked by an ‘infectiologist’?
2. Discussion – lines 372-380 – should be merged into the Introduction and not appear here. Limitations - can also include non-standardised evaluation of NS and not by neurologist. Note the NS is not only cognition eg numbness, weakness, etc are NS??
